# Review of Single-Cell RNA Sequencing in the Heart

**DOI:** 10.3390/ijms21218345

**Published:** 2020-11-06

**Authors:** Shintaro Yamada, Seitaro Nomura

**Affiliations:** Department of Cardiovascular Medicine, Graduate School of Medicine, The University of Tokyo, Tokyo 113-8654, Japan; shintayamada-tky@umin.ac.jp

**Keywords:** single-cell RNA sequencing, heart, cardiomyocyte, cardiovascular development, cardiovascular disease

## Abstract

Single-cell RNA sequencing (scRNA-seq) technology is a powerful, rapidly developing tool for characterizing individual cells and elucidating biological mechanisms at the cellular level. Cardiovascular disease is one of the major causes of death worldwide and its precise pathology remains unclear. scRNA-seq has provided many novel insights into both healthy and pathological hearts. In this review, we summarize the various scRNA-seq platforms and describe the molecular mechanisms of cardiovascular development and disease revealed by scRNA-seq analysis. We then describe the latest technological advances in scRNA-seq. Finally, we discuss how to translate basic research into clinical medicine using scRNA-seq technology.

## 1. Introduction

To understand the phenomenon of life, it is important to elucidate the biological mechanisms of the cells that constitute living organisms. RNA is essential to biological processes in cells, and transcriptomes provide critical information directly associated with cell phenotypes. Single-cell RNA sequencing (scRNA-seq) is a powerful tool for characterizing individual cells. The conventional technique of bulk RNA-seq measures the average gene expression across cells in a sample and identifies differences between sample conditions, whereas scRNA-seq measures the gene expression of individual cells and can identify differences between cells in one or more samples. Although cells are traditionally characterized morphologically or by molecules unique to each cell type, scRNA-seq facilitates automatic classification of cells via clustering of transcriptomes and can identify heterogeneous cell types and molecular states even in a group that have been considered to consist of only one cell type [1].

Cardiovascular disease is common worldwide. One in a hundred children have congenital heart disease, such as ventricular septal defect and tetralogy of Fallot, resulting from an abnormality during the heart development process. For adults, cardiovascular disease is one of the major causes of death worldwide and includes cardiomyopathy, ischemic heart disease, and valvular disease, which can lead to heart failure and even sudden death. The precise pathology remains to be elucidated because of the complexity of interactions between cells and molecules not only within the heart but also across other organs.

The heart is exposed to various types of stress. Cardiomyocytes (CMs) have an abundant supply of mitochondria, which not only allows them to produce energy for contraction but also generates reactive oxygen species and leads to cellular toxicity [2]. The heart constantly contracts and is under high pressure from the blood, especially in the left ventricle, which can cause mechanical stress and hypertrophy [3]. Several molecules involved in the renin–angiotensin system and the sympathetic nervous system and cytokines affect the function of the heart.

In past decades, conventional molecular biological studies have partly revealed the pathological mechanism underlying cardiovascular disease, but further studies are warranted. The development of scRNA-seq techniques could provide new insights into both healthy and pathological hearts.

The content of this review is summarized in Figure 1. We first introduce the scRNA-seq platforms appropriate for each cell type. Next, we describe the molecular mechanisms of cardiovascular development and disease revealed by scRNA-seq analysis. Then, we present the latest technological advances in scRNA-seq. Finally, we discuss how to translate basic research into clinical medicine using scRNA-seq technology.

## 2. scRNA-seq Platforms

### 2.1. Platforms for Small Cells

Dispensing tissues into a single cell is an important process in scRNA-seq analysis. Fluorescence-activated cell sorting (FACS) is conventionally used for single-cell sorting [4] and utilizes scatter and fluorescence signals to sort cells into 96- or 384-well plates. The use of fluorescence signals enables the filtering of dead cells and the selection of cells of interest. Because of the size of the nozzle, cells less than 50 μm in diameter can be sorted with minimal damage, but it is difficult to sort adult CMs due to their large size (>100 μm in diameter along the major axis). Another option is to use the integrated fluidic circuit (IFC) system (e.g., Fluidigm C1), which isolates single cells into individual reaction chambers in a more automated manner. Cells are automatically lysed, and cDNA libraries are quickly prepared for sequencing. IFC captures cells less than 25 μm in diameter and processes up to 800 cells. FACS and IFC are plate-based platforms, and thus the number of cells that can be sorted and analyzed is limited.

Both Drop-seq [5] and Chromium (10× Genomics), which is a commercial droplet-based platform, allow for the rapid profiling of thousands of individual cells by encapsulating them in tiny droplets, adding barcodes to each cell’s RNAs, and sequencing them together. These approaches utilize a microfluidics device to perform droplet separation; therefore, the size of cells that can be analyzed is generally less than 30 μm in diameter.

The droplet-based system is superior to the plate-based system in terms of throughput, but the former tends to be less effective at detecting genes due to its low capture rate of mRNA molecules and it has a higher tendency to generate doublets. Most studies using a droplet system detected around 500–1500 genes, whereas those using a plate-based system detected around 1000–4000 genes. It is important to select methods according to how many cells will be analyzed and how many genes are expected to be detected.

These plate-based (FACS and IFC system) and droplet-based platforms (Drop-seq and Chromium) are suitable for scRNA-seq of normal and small cells. Although adult CMs in mice and humans are relatively large, embryonic and neonatal CMs can be analyzed using such systems. In addition, non-CMs, such as fibroblasts, endothelial cells, and macrophages, and nuclei extracted from adult CMs are small enough to be analyzed using these systems.

### 2.2. Platforms for Large Cells

The channel diameter of the conventional nozzle in FACS, IFC, and droplet-based systems is too narrow for adult CMs to pass through without damage. Gladka et al. used FACS with a larger nozzle size (130 μm) to perform scRNA-seq of adult murine CMs [6]. ICELL8 is another promising platform that uses a large-bore nozzle dispenser to distribute single cells from diluted cell suspensions into 5184 nanowells. ICELL8 also has an imaging system that visualizes all nanowells. Using fluorescence signals of the imaging system makes it possible to differentiate live cells from dead cells, or wells containing a single cell from wells containing no or multiple cells, generating 1000–1500 single-cell transcriptomes. Manual pick-up and mouth pipettes are also utilized in the scRNA-seq of adult CMs because they are gentler on cells in terms of stress compared with FACS. Using these strategies, we can use a microscope to choose the cardiomyocytes with the best shape for sequencing. However, some training is necessary for manual sorting of individual cells and the throughput is lower.

There are few platforms for scRNA-seq of adult CMs, and given that all of them are plate-based systems, the number of cells they can analyze is limited. Therefore, some researchers have used nuclei extracted from adult CMs, rather than intact CMs, in order to utilize the droplet-based system.

## 3. scRNA-seq and Cardiovascular Development and Disease

### 3.1. Murine Embryonic and Neonatal Heart

scRNA-seq researches regarding cardiovascular development and disease are summarized in Table 1. Compared with adult hearts, it is easier to digest embryonic and neonatal hearts, isolate their CMs, and sort them because their extracellular matrices are softer, they have higher viability ex vivo, and their CMs are smaller. Consequently, healthy/pathological differentiation and developmental processes have been well studied in embryonic and neonatal murine hearts. scRNA-seq of CMs from murine embryonic or neonatal hearts have been performed using IFC, FACS, and a mouth pipette or droplet systems.

scRNA-seq has revealed some key factors in CM maturation. Two groups applied an IFC system to obtain the transcriptomes of thousands of cells from embryonic and postnatal hearts [7,8]. They revealed spatiotemporal transcriptomic changes during heart development and showed that Nkx2-5 is an important factor in CM maturation. Xiong et al. utilized FACS and revealed that Nkx2-5 directly regulates spatial expression of the chemokine receptors Cxcr2 and Cxcr4 in the second heart field, which directs second heart field cardiac progenitor migration with spatiotemporal precision [9]. In addition to Nkx2-5, Mesp1 is also essential for CM maturation. scRNA-seq of embryonic cardiac progenitors has revealed that Mesp1 is required for exit from the pluripotent state and induction of the cardiovascular gene expression program [10].

scRNA-seq has furthermore revealed that heart development does not take place only within cardiomyocytes. scRNA-seq of various tissues and organs from embryonic mice identified mutual interactions between epithelial and mesenchymal cells [11]. Epithelial cells with prevalent mesenchymal features were detected during organogenesis, and had similar features to those of intermediate epithelial/mesenchymal cells during tumorigenesis. As mentioned above, IFC system, FACS, and mouth pipette systems have all been utilized for scRNA-seq of the embryonic or neonatal murine heart and have revealed critical factors for cardiogenesis. However, the number of cells that can be sorted and analyzed is limited to less than a few thousand because they are all plate-based methods.

Droplet systems are a high-throughput scRNA-seq methods and have revealed other critical factors for CM maturation. scRNA-seq of 73,926 cells from embryonic murine hearts was performed using Chromium [12], and Hand2 was identified as a specifier of outflow tract cells but not right ventricular cells, although *Hand2* knockout mice are known to show failure of right ventricular formation. The finding that Hand2 expression is regulated by non-coding RNA Hand2os1 was also demonstrated by scRNA-sq using Chromium [13]. scRNA-seq is used to investigate not only healthy but also pathological development. The transcriptome of 15,083 nuclei from healthy embryonic mouse hearts and from the hearts of a murine model of pediatric mitochondrial cardiomyopathy was analyzed using Chromium [14]. This study revealed cell type-specific differences of the transcriptional landscapes, including changes of subtype composition, maturation states, and functional remodeling of each cell type and found transcriptional activation of Gdf15 in the disease model.

Recently, specific regions of the heart have been studied using scRNA-seq: the cardiac conduction system in embryonic mouse [15], aortic and mitral heart valves in postnatal mice [16], and the cardiac outflow tract in embryonic mice [17]. All of these procedures were performed using a droplet-based system and the transcriptomes of 2840–55611 cells were analyzed. Consequently, uniquely expressed genes in each region and changes in expression during heart development were identified.

scRNA-seq of embryonic and neonatal murine hearts have revealed some key factors in CM maturation such as Nkx2-5, Mesp1, and Hand2. During cardiac development, metabolic reprogramming, withdrawal from cell-cycle, establishment of cell–cell interaction, and multinucleation simultaneously occurred in cardiomyocytes. scRNA-seq analysis research addressing these complex questions are expected.

### 3.2. Murine Adult Heart

Although embryonic and neonatal hearts are used to study cardiac development and differentiation, adult hearts are often used to investigate pathogenesis in murine disease models. Three disease models are commonly used: myocardial infarction (MI) with occlusion of the left anterior descending artery (LAD); ischemia/reperfusion (I/R) with LAD occlusion followed by LAD reperfusion; and pressure overload with transverse aortic constriction (TAC). Adult CMs are vulnerable to shock, sensitive to pH and calcium concentration, and are relatively large (>100 μm in diameter along the major axis); therefore, they are more difficult to isolate and sort compared with embryonic or neonatal CMs [39]. Some platforms have been improved to allow scRNA-seq of adult CM, and they have revealed some key mechanisms and responses in disease models.

Ischemic heart disease such as MI and angina pectoris is one of the most common cardiac diseases. Understanding the molecular mechanisms of cellular responses after ischemic stress has the potential to lead to the development of patient stratification methods and novel therapeutic approaches. scRNA-seq of cardiac cells from MI and I/R models by FACS using a large nozzle size (130 μm) identified an activated fibroblast marker but found no evidence for proliferation of significant numbers of CMs in response to cardiac injury [6,19]. These findings suggest the possibility of preventing excessive fibrosis after MI by controlling fibroblast activation and the difficulty of accelerating CM proliferation.

TAC, which is another popular disease model, induces pressure overload on the heart. The heart incessantly contracts and is under high pressure from the blood, especially in the left ventricle, which can cause mechanical stress and hypertrophy [3]; therefore, understanding the association of pressure overload and heart function could lead to novel insights into the molecular mechanisms of heart failure. scRNA-seq of cells from a TAC mouse model by using ICELL8 showed heterogenous transcriptional signatures among CMs and revealed that macrophage activation was a key factor during cardiac hypertrophy [18,20]. Nomura et al. manually isolated single cardiomyocytes from wild-type and *p53* cardiomyocyte-specific knockout mice with healthy and TAC model [22]. They analyzed transcriptomes of 473 CMs, and found that persistent overload leads to a bifurcation into adaptive and failing CMs, and that p53 signaling is specifically activated in late hypertrophy. They also manually isolated and analyzed 411 adult human CMs from healthy individuals and patients with dilated cardiomyopathy, and validated the conservation of pathological transcriptional signatures. Another report using ICELL8 demonstrated that cardiac fibroblasts are a key constituent in the microenvironment and promote CM maturation [21]. Although it remains fully elusive what causes heterogenous transcriptional signatures among CMs, intrinsic (e.g., p53) and extrinsic signals (from macrophages and/or fibroblasts) may induce the transition into failing CMs. Given that adult CMs are too large for use with a droplet-based system, researchers performed scRNA-seq using FACS with a large nozzle size, ICELL8, or manual isolation to demonstrate heterogeneous changes in gene expression after TAC, MI, or I/R procedure.

Intact adult CMs are too large to apply to a droplet system, and thus some researchers have extracted CM nuclei for use with this system to perform high-throughput scRNA-seq. Single-nucleus RNA-seq (snRNA-seq) of tens of thousands of nuclei extracted from healthy mice and from a murine MI model were performed using Chromium [23,24]. The results revealed that dedifferentiation may be an important prerequisite for CM proliferation and demonstrated the limited but measurable cardiac myogenesis after MI. In addition, fibroblasts were shown to develop diverse transcription profiles with aging. Linscheid et al. focused on cells from the sinus node [26]. They performed snRNA-seq of 5357 nuclei from the sinus node of healthy adult mice. They detected myocytes, fibroblasts, endocardial cells, epicardial cells, epithelial cells, adipocytes, macrophages, neurons, and vascular endothelial cells in the sinus node and revealed that membrane clock proteins were predominantly expressed in the sinus node myocytes.

Adult CMs are difficult to apply to a droplet system, whereas non-CMs can be easily applied because of their size. scRNA-seq of tens of thousands of cardiac interstitial cells after a MI procedure were performed using Chromium [27,28]. The results revealed an activated fibroblast population and their marker as well as a subpopulation of resident endothelial cells, which contribute to new blood vessel formation following MI.

In summary, FACS with a large nozzle size, ICELL8, and a manual isolation have all been utilized to dispense adult mouse CMs. However, the number of cells that can be analyzed using these methods is limited. A droplet system has a higher throughput, but cannot accommodate the large size of intact CMs. Therefore, nuclei extracted from CMs and non-CMs have been studied using droplet systems. Both scRNA-seq and snRNA-seq have revealed some key factors and mechanisms related to the pathological heart after pressure overload and ischemia.

### 3.3. Human-Induced Pluripotent Stem Cells

Human induced pluripotent stem cells (hiPSCs) are a pivotal biological model [40]. Once hiPSCs are established, they are considered to have an infinite self-renewal capacity. Although it is difficult to collect large enough samples from human hearts, sufficient amounts of hiPSCs can be easily prepared. Because they contain information from the most basic levels of the human genome, we can perform scRNA-seq to infer the molecular behaviors in the development and pathogenesis of human hearts. hiPSCs can be isolated in vitro; therefore, it is easier to isolate and dispense cardiomyocytes for scRNA-seq compared with in vivo procedures.

scRNA-seq of hiPSCs have revealed some key factors in cardiac differentiation in humans. scRNA-seq of 43,168 hiPSCs during cardiac differentiation was performed using Chromium [29] and demonstrated that hypertrophic signaling is not sufficiently activated during monolayer-based cardiac differentiation, thereby preventing expression of HOPX and its activation of downstream genes governing the late stages of CM maturation. Other studies of scRNA-seq performed using Chromium, IFC, or FACS identified temporal and spatial gene expression changes and their key factors such as ISL1, NR2F2, TBX5, and HEY2 during cardiac differentiation [30,33,34].

iPS cell researches are anticipated to be helpful for the development of cardiac regeneration therapy and for the understanding of the disease pathogenesis. scRNA-seq analysis will expand these possibilities.

### 3.4. Human Heart

It is preferable to use human cells for human disease research. However, it is difficult to collect samples. Patients with heart disease occasionally undergo biopsy, but the amount of tissue retrieved is insufficient. In addition, healthy individuals do not undergo biopsy; therefore, it is almost impossible to collect healthy living heart tissue. Instead, healthy hearts from deceased individuals are used as controls.

Like embryonic mouse CMs, embryonic human CMs are small enough for use with any scRNA-seq platform. scRNA-seq of these CMs has detected some key factors in both healthy and pathological heart development. scRNA-seq of embryonic human CMs with a mouth pipette and FACS identified region-specific markers early in cardiac development, including *LGR5*^+^ cardiac progenitor cells in the proximal outflow tract [31,35]. Asp et al. used Chromium to analyze 4026 single-cell transcriptomes from embryonic hearts [36]. They also performed RNA-seq using spatial transcriptomics and in situ sequencing. By combining their data, they generated a 3D gene expression atlas of the developing human heart. Cardiac cells from healthy fetal hearts and fetal hearts affected by autoimmune-associated congenital heart block were collected and their transcriptome was analyzed [37]. The results showed increased and heterogeneous interferon responses in various cardiac cell types with congenital heart block compared with the healthy control.

As is the case with adult mouse CMs, the methods for sequencing adult human CMs are limited because of their size. As mentioned above, Nomura et al. manually isolated individual mouse and human CMs [22]. They integrated gene expression profiles with multidimensional data to dissect the molecular and morphological dynamics of CMs during cardiac hypertrophy and heart failure through co-expression network analysis of mouse and human single-CM transcriptomes. Wang et al. used ICELL8 to analyze 21,422 single-cell transcriptomes from healthy, failed, and partially recovered adult human hearts. They revealed cellular heterogeneity and found that CM contraction and metabolism are the two most critical aspects affected by heart function deterioration [38].

Intact adult CMs are too large to apply to a droplet-based system; therefore, CM nuclei have been used in droplet-based systems, which have a higher throughput and can be used to analyze frozen specimens. In addition, extracting the nuclei can also remove mitochondria, which are abundant in CMs. Selewa et al. compared snRNA-seq with scRNA-seq in terms of read depth, transcriptome composition, cell types detected, and cellular differentiation trajectories [32]. They adapted Drop-seq to both the cells and nuclei (DropNuc-seq) of hiPSCs and confirmed that inclusion of reads from intronic regions increases the sensitivity of transcriptomes from nuclei and improves the resolution of cell type identification. Then, they applied DropNuc-seq of single nuclei to frozen postmortem human heart tissue and demonstrated that DropNuc-seq has the ability to detect cardiac cell types. See et al. utilized the IFC system and analyzed 359 CM nuclei from mouse and human left ventricles [25]. They found that subpopulations of CMs upregulate cell-cycle activators and inhibitors as a consequence of the stress-response, thereby discovering long intergenic non-coding RNAs as key nodal regulators.

When fresh adult human heart tissue is available, ICELL8 or manual isolation is an appropriate platform for performing scRNA-seq. When only frozen heart tissue is available, snRNA-seq using a droplet-based system is suitable. Although studies of the human heart have been conducted as described above, they are fewer in number compared with murine heart studies due to the difficulty in collecting sufficient amounts of human heart tissue. Accordingly, the study of mechanisms underlying cardiac development and disease in human hearts through the use of scRNA-seq is ongoing.

## 4. scRNA-seq and Applications

### 4.1. Cell–Cell Interactions

scRNA-seq is a powerful tool for elucidating biological mechanisms at the cellular level, but many organisms do not consist of a single cell. Therefore, it is important to understand the interactions between cells. There have been some studies regarding cell–cell interaction inferred by scRNA-seq in organs other than the heart.

Cohen et al. analyzed ligand–receptor interactions from the lung transcriptomes of about 50,000 single cells [41]. They found that basophils are important regulators in lung tissue. Basophils receive interleukin (IL)-33 from alveolar type 2 cells and then secrete IL-6 and IL-13 to promote maturation of alveolar macrophages. Vento-Tormo et al. profiled the transcriptomes of about 70,000 single cells from first-trimester placentas with matched maternal blood and decidual cells, and identified three types of natural killer cells [42]. They also specified a distinctive chemokine profile for each natural killer cell and cell–cell communication with the new repository CellPhoneDB.

Cell–cell communication has been confirmed not only within but also among tissues. Ma et al. built comprehensive single-cell and single-nucleus transcriptomic atlases across various rat tissues (fat, aorta, kidney, liver, skin, bone marrow, muscle, and brain) during aging and caloric restriction [43]. They found that the numbers of cells associated with the immune and inflammatory systems increased with aging and were suppressed with caloric restriction among various tissues. Excessive proinflammatory ligand–receptor interplay was observed during aging, but this was reversed by caloric restriction.

In the heart, ligand–receptor analysis revealed interactions between cardiac cell types in homeostasis and injury [23,27]. In particular, there is a strong association between fibroblasts and endothelial cells after injury or during aging. Although there are few scRNA-seq studies of cell–cell interactions in the heart, it is known that these interactions are important not only within heart tissue but also among several organs involved in MI and heart failure. The activation of macrophages and cardiac inflammation after MI are associated with long-term cardiac function. In addition, a decrease in the brain’s gray matter is related to heart failure [44,45]. Better understanding of communication between cells by single-cell analysis would lead to increased understanding of mechanisms involved in heart development and disease.

### 4.2. Spatial Transcriptomes

Spatial information is an important element in single-cell analysis. Lubeck et al. developed sequential fluorescence in situ hybridization (seqFISH) and Eng et al. further developed this method as seqFISH+ [46,47]. In seqFISH+, transcripts are labeled with fluorescent probes in sequential rounds of hybridization to read out the temporal barcode for each transcript and this can image mRNAs for 10,000 genes in single cells with a high accuracy. Co-Detection by Indexing (CODEX) quantifies protein expression at the single-cell level based on antigen–antibody immunohistochemical interaction, whereas seqFISH+ quantifies gene expression at the single-cell level based on mRNA in situ hybridization technology [48]. CODEX utilizes hybridization of the DNA tag of oligonucleotide-conjugated antibody and fluorophore-labeled nucleotides. Goltsev et al. applied CODEX to mouse spleens and observed an unexpected profound impact of the cellular neighborhood on the expression of protein receptors on immune cells [48]. Jackson et al. used imaging mass cytometry (Hyperion Imaging System by Fluidigm, South San Francisco, CA, USA) to analyze the localization of 35 proteins in 171,288 cells from 281 breast cancer patients [49]. This spatially resolved, single-cell analysis revealed that multicellular features of the tumor microenvironment are associated with distinct clinical outcomes.

Some studies have investigated spatial gene expression in the heart. For example, Satoh et al. spatially quantified gene expression at the single-cell level in the heart after TAC and found spatially heterogenous *Myh7* expression in CMs after TAC [50]. Asp et al. combined scRNA-seq data of human embryonic cardiac cells, RNA-seq data of spatial transcriptomics, and in situ sequencing data to generate a 3D gene expression atlas of the developing human heart [36]. Although there are few reports on spatial transcriptomics in the heart, cardiac development is a spatially complex process and regional changes in gene expression during heart maturation is a process of great interest. Spatial information not only in the developing heart but also in the adult heart, particularly in MI, is important. Given that MI is caused by occlusion of a coronary artery, the infarct site is directly affected by ischemia, whereas the opposite site is indirectly affected. It is known that functional neovascularization in the infarct border may delay progression to heart failure and improve the outcome [28]. Accordingly, understanding transcriptomic changes in each region would lead to new therapeutic targets for MI.

### 4.3. Trajectory Analysis

Temporal evaluation is also another important element in single-cell analysis. Temporal changes of gene expression are essential during development or disease progression. There are two main methods to analyze temporal gene expression changes: prediction of the cell trajectory based on scRNA-seq (known as trajectory inference); and utilizing lineage of DNA barcodes and integrating the transcriptomes of each cell (lineage tracing).

Trajectory inference interprets a single cell as a snapshot of a continuous process and reconstructs the pseudotime by analyzing transcriptional changes between neighboring cells [51]. Several trajectory inference methods, such as Monocle [52], Wanderlust [53], Slingshot [54], and PAGA [55], have been developed. These methods differ in the complexity of the model paths which are simple linear, bifurcating, or multifurcating trajectories. Velocyto is a package for the analysis of expression dynamics from the estimation of RNA velocities by distinguishing unspliced and spliced mRNAs [56]. They have been used in studies of cardiac development and disease progression [9,11,14,21,22,23,32,36,38].

Mother and daughter cells have essentially the same genomic DNA. Lineage tracing can identify which cells are generated from a particular cell by adding barcodes to the genomic DNA of the mother cells before division. There are several methods for inserting barcodes to genomic DNA, including applying a recombination using the Cre-loxP system [57], insertion by lentivirus [58,59], or using double-stand breaks of DNA by CRISPR-Cas9 [60,61,62]. Some CRISPR-Cas9 barcoding methods are designed to express barcodes as mRNA, which enables a simultaneous analysis of the DNA barcodes and transcriptomes in single cells, but most methods depend on double-stand breaks; therefore, they might affect development or disease progression.

Although trajectory inference has been used in studies of the embryonic heart and revealed some important mechanisms in heart development, to date, no scRNA-seq studies have been conducted using lineage tracing with barcodes. Lineage tracing can clarify where the cell is descended from and might provide greater insight into the development of the heart. Although adult hearts have little proliferative activity [63], epicardial cells undergo epithelial-to-mesenchymal transition after MI [64]. Lineage tracing using scRNA-seq analysis has the potential to reveal the precise mechanisms of a cellular transition in the heart.

## 5. Future Perspectives

The development of scRNA-seq techniques have provided some novel perspectives on both healthy and pathological hearts. scRNA-seq have revealed several important molecular mechanisms in cardiac development and pathogenesis. Furthermore, this novel technology has been applied to human pathogenesis. Now, scRNA-seq analysis researches are advancing by using molecular information not only within a single cell but also across various cells and organs with high spatial and temporal accuracy.

In addition, the linking of scRNA-seq and clinical data from other organs has been proceeding. Velmeshev et al. performed snRNA-seq of cortical tissue from individuals with autism [65]. They found that specific sets of genes in upper-layer projection neurons and microglia correlate with the clinical severity of autism. Exome sequencing was used to confirm these as causative genes [66]. Kim et al. performed scRNA-seq on skin and blood from a patient with refractory drug-induced hypersensitivity syndrome/drug reaction with eosinophilia and systemic symptoms [67]. They identified the JAK-STAT signaling pathway as a potential target and showed that tofacitinib, a JAK1 and JAK3 inhibitor, was effective in controlling disease. Yamaguchi et al. conducted scRNA-seq of isolated cardiomyocytes from heart failure patients with ventricular arrythmia and identified dopamine D1 receptor (D1R)-expressing cardiomyocytes [68]. They generated cardiomyocyte-specific *D1R* knockout and transgenic (overexpression) mice to show that the upregulation of cardiac D1R is both necessary and sufficient for triggering life-threatening ventricular arrythmia by driving dysregulated Ca^2+^ handling via hyperactivation of ryanodine receptor 2 in the failing heart, independent of cardiac function.

As with these other organs, linking the scRNA-seq and clinical data of hearts has the potential to lead to novel insights and treatments. To achieve this aim, it is essential to translate basic research into clinical medicine.

## Figures and Tables

**Figure 1 ijms-21-08345-f001:**
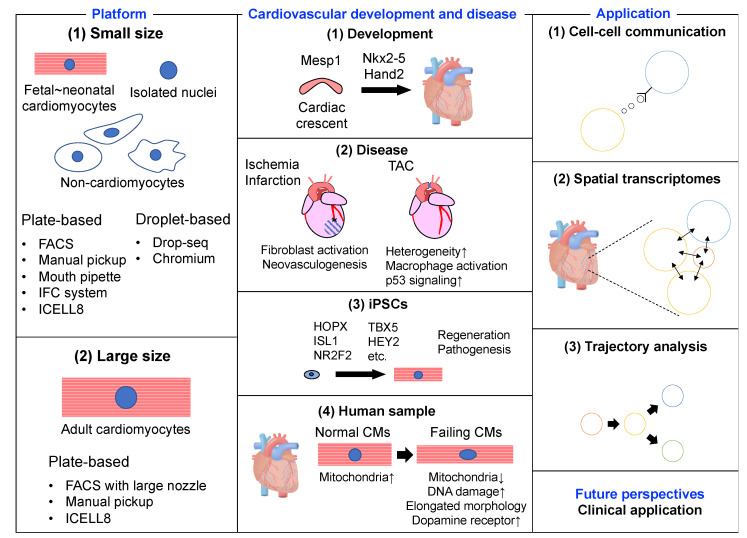
Summary figure of single-cell RNA-sequencing in the heart.

**Table 1 ijms-21-08345-t001:** Single-cell RNA sequencing in the heart.

Species	Vivo / Vitro	Age	Cells or Nuclei	Model	Device	Number of Cells for Analysis	Findings	Ref.
mouse	vivo	fetus	cells from whole heart	healthy development	IFC system	2233 cells	chamber-specific genes in the embryonic mouse heart	[8]
		fetus	cells from whole heart	healthy development and *Hand2* knock out	Chromium	73,926 cells	Hand2 is a specifier of outflow tract cells	[12]
		fetus	cells from whole heart	healthy development and *Hand2os1* knock out	Chromium	3600 cells	lncRNA Hand2os1/Uph regulates Hand2	[13]
		fetus	cells from whole heart and other 7 organs	healthy development	mouth pipette	1819 cells	mutual interactions between epithelial and mesenchymal cells	[11]
		fetus	*Mesp1* positive or null cardiac progenitors	healthy development and *Mesp1* knock out	FACS	598cells	Mesp1 is required for the exit from the pluripotent state	[10]
		fetus	*Nkx2.5* or *Isl1* expressing cardiac progenitors	healthy development	FACS	1231 cells	Cxcr2 regulates chemotaxis during development	[9]
		fetus	cells from cardiac conduction system	healthy development	Chromium	22,462 cells	transcriptional profiles of cardiac conduction system	[15]
		fetus	cells from cardiac outflow tract	healthy development	Chromium	55,611cells	cellular transitions in cardiac outflow tract	[17]
		fetus ~ neonate	cells from whole heart	healthy development	IFC system	>1200 cells	temporal and chamber-specific markers during development	[7]
		neonate	nuclei from whole heart	healthy development and pediatric mitochondrial cardiomyopathy	Chromium	15,083 nuclei	heterogeneity of various cell types	[14]
		neonate	cells from left ventricles	healthy development	ICELL8	4231 cells	transcriptomes of mono- or multi-nucleated cardiomyocytes are highly similar	[18]
		neonate ~ juvenile	cells from aortic valve and mitral valve	healthy development	Drop-seq	2840 cells	Interstitial cell subpopulations undergo changes in gene expression during development	[16]
		neonate, adult	cells from ventricles	healthy, I/R and MI	FACS	1939 cells	Cycling CMs are few adult mouse	[19]
mouse	vivo	adult	cells from whole heart	healthy condition and ischemia reperfusion	FACS	935 cells	Ckap4 is a modulator of fibroblasts activation	[6]
		adult	cells from whole heart	healthy and TAC	ICELL8	11,492 cells	Macrophage activation is a key factor of hypertrophy	[20]
		adult	cells from left ventricles	healthy development	ICELL8	2497 cells	Fibroblast regulates CM maturation	[21]
		adult	CMs from ventricles	healthy and TAC	ICELL 8	<1015 cells	heterogeneity among CMs after TAC	[18]
		adult	CMs from whole heart	healthy and TAC	manual pickup	396 cells	p53 induces molecular and morphological remodeling	[22]
		adult	nuclei from whole heart	healthy aging	Chromium	27,808 nuclei	heterogeneity of fibroblasts with aging	[23]
		adult	nuclei from ventricles	healthy and MI	Chromium	31,542 nuclei	dedifferentiation in cycling CMs after MI	[24]
		adult	nuclei of CMs from left ventricles	healthy and TAC	IFC system	243 nuclei	lincRNA regulates dedifferentiation and cell cycle genes	[25]
		adult	cells from sinus node	healthy pacemaking	Chromium	5357 nuclei	Membrane clock underpins pacemaking	[26]
		adult	non-CMs	healthy and MI	Chromium	13,331 cells	transcriptome changes of non-CMs after MI	[27]
		adult	fibroblasts	healthy and MI	IFC system	104 cells	transcriptome changes of fibroblast after MI	[27]
		adult	endothelial cells	healthy and MI	Chromium	28,598 cells	Plvap regulates endothelial proliferation	[28]
		neonate, adult	neonatal CMs, and neonatal and adult fibroblasts	healthy development	ICELL8	1580 cells	Fibroblast regulates CM maturation	[21]
human	vitro		hiPSC-CMs	differentiation	Chromium	43,168 cells	Hopx is a key regulator of CM maturation	[29]
			hiPSC-CMs	differentiation	Chromium	10,376 cells	ISL1, NR2F2, TBX5, HEY2, or HOPX are makers of hiPSC-CMs	[30]
			hiPSC-CMs	differentiation	IFC system	43 cells	ISL1, NR2F2, TBX5, HEY2, or HOPX are makers of hiPSC-CMs	[30]
			CMs derived from embryonic stem cells	differentiation	FACS	366 cells	LGR5 is a marker of cardiac progenitors in embryonic outflow tract	[31]
			hiPSC-CMs	differentiation	Drop-seq	23,554 cells	the comparison with DroNc-seq	[32]
			nuclei of hiPSC-CMs	differentiation	DroNc-seq	24,318 nuclei	Inclusion of reads from intronic regions increases the sensitivity	[32]
			epicardium hiPSC-CMs	differentiation	FACS	232 cells	BNC1 regulates cell heterogeneity	[33]
			CMs reprogrammed from human fibroblasts	differentiation	IFC system	704 cells	cell fate transitions during reprogramming	[34]
human	vivo	fetus	cells from free wall	healthy development	mouth pipette	3842 cells	Atrial and ventricular CMs acquires distinct features early in heart development	[35]
		fetus	cells from whole heart	healthy development	Chromium	4026 cells	cell atlas of the developing human heart	[36]
		fetus	cells from whole heart	healthy development	FACS	458 cells	LGR5 is a marker of cardiac progenitors in embryonic outflow tract	[31]
		fetus	cells from whole heart	healthy and autoimmune-associated congenital heart block	Chromium	17,747 cells	heterogeneous interferon responses in congenital heart block heart	[37]
		adult	cells from whole heart	healthy, HF and functional recovery from HF after treatment with LVAD	ICELL8	21,422 cells	CM contractility and metabolism are prominent aspects that are correlated with changes in heart function.	[38]
		adult	CMs from left ventricles	healthy and DCM	manual pickup	411 cells	heterogeneity in DCM CMs	[22]
		adult	nuclei from whole heart	healthy	DroNuc-seq	1491 nuclei	the usefulness of DroNc-seq in adult human CMs	[32]
		adult	nuclei from CMs	healthy and DCM	IFC system	116 nuclei	lincRNA regulates dedifferentiation and cell cycle genes	[25]

CMs, cardiomyocytes; hiPSC-CMs, human induced pluripotent stem cells-derived cardiomyocytes; I/R, ishchemia/reperfusion; DCM, dilated cardiomyopathy; MI, myocardial infarction; TAC, transverse aortic constriction; HF, heart failure; LVAD, left ventricular assist device; FACS, fluorescence-activated cell sorting; IFC, integrated fluidic circuit.

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
