# Peer review of "Review of Single-Cell RNA Sequencing in the Heart"

_ijms, 2020, doi:10.3390/ijms21218345_

Round 1
Reviewer 1 Report
In this manuscript, Shintaro Yamada and Seitaro Nomura specifically summarized the applications of scRNA-seq in cardiovascular field. It has covered most recent studies that used scRNA-seq to address biology questions related to heart development and diseases. Further, the authors discussed potential translation of scRNA-seq technology to clinical applications. Overall, this is an interesting review that may offer useful information and provide insights into future use of scRNA-seq technology for cardiovascular researches. Major comments:
- The authors provide general information on the different platforms of scRNA-seq. Since the authors aimed to discuss the application of scRNA-seq, a detailed section describing the advantages and disadvantages of each platform would make the manuscript more informative, benefit readers for understanding why specific platform is chosen in context-specific studies , and help to decide which platform would be used.
- The authors covered the broad advances of scRNAseq in the process of heart development and cardiovascular diseases. It would be better if the authors could summarize the main findings from the literature, get conclusion and then cite the reference that support the conclusion. Sections in current manuscript seemed to be independent from each other and no internal connection could be identified, which reduce reviewer’s interest.
- The authors may also improve their writing. Current format is kind of descriptive and needs further in-depth summary.
- Transitions inside each section should be improved. For example, in 3.2 section, it is hard to connect these paragraphs to obtain a consensus conclusion.
- There are many grammatical errors in the article, which affect the reviewer's reading experience.
Minor issue:
- In 2.1, references [6] seem unnecessary.
- Figure 1 needs further refinement.
Reviewer 2 Report
The review entitled “Review of single-cell RNA sequencing in the heart”” gives an overview on the state-of-the-art about the single-cell RNA approaches for the study of cardiovascular disease. I think that the topic is extremely interesting and that the manuscript is quite clear and precise, however, there are some points the need to be addressed to render the paper suitable for the publication.
- In general, all the paragraphs present a punctual list of the studies published. I think that a review should give a critical view of the state-of-the-art of a particular matter, and not only report the results from other papers. I invite the authors to add, at the end of each paragraph, their point of view of the results described, not limited to the technical aspect.
- The authors should clarify what they mean talking about “cardiovascular disease”. I think that “cardiovascular disease” is too reductive and simplistic. Indeed, cardiovascular diseases comprise a wide range of pathologies, including for example cardiomyopathies, arrhythmias, vascular diseases, congenital malformations. The lanes 30-38 should be re-written to better define this aspect.
- I think that would be extremely interesting to add a figure to paragraph 2.1, combining all the results obtained in the different studies reported. The resulting image could be a flowchart of heart development, highlighting the timing and the role of the different genes identified.
- Also in the “scRNA-seq and application” section, there is a list of the published work without an opinion. Moreover, the part regarding the heart is very limited, as compared to the other works. Please, improve these aspects
- The “Future perspective” paragraph presents the same problem. In particular, this section must represent the section where the authors must give their point of view on the topic that they have presented, highlighting the pro and cons, and suggesting the future developments.
- I suggest adding a section explaining the methods used for the scRNA (i.e. FACS, manual pick-up, ICELL8, Chromium, nuclei isolation), evidencing the technical differences among them.
- Figure 1 is not mentioned in the text.
Minor revisions:
- Add some references at lane 55.
Add some references at lanes 175-177.
Round 2
Reviewer 2 Report
I thank the author to address all my previous comments. I think that the manuscript has been improved and now it is suitable for publication.